**Data Availability Statement:** Data cannot be shared publicly because of ethical restrictions due to data containing potentially identifying or

# Determinants of mortality among preterm neonates admitted at Wolaita Sodo University Comprehensive Specialized Hospital, southern Ethiopia: An *unmatched case-control study*

**Alemu Bogale**[1]*, **Ushula Deboch Borko**[1], **Solomon Abreha**[2], **Bargude Balta**[3]

**1** School of Medicine, College of Health Science and Medicine, Wolaita Sodo University, Wolaita Sodo, Ethiopia, **2** School of Public Health, College of Health Science and Medicine, Wolaita Sodo University, Wolaita Sodo, Ethiopia, **3** Department of Nursing, Hawassa University Comprehensive Specialized Hospital, Hawassa, Ethiopia

* alemubogale93@gmail.com

## Abstract

### Background

Preterm birth accounts about 1 million neonatal deaths worldwide and the second causes of both neonatal and under five-child mortality. From this preterm is the second leading cause of death and is steadily increased in low-income countries. In Ethiopia, Preterm neonatal death is the first and fourth cause of newborn and under-5 deaths, respectively. Although the general newborn mortality rate in the research area was 27 per 1000 neonates, but the preterm neonatal mortality rate and determinants were not previously well recognized. This study aimed to identify the Determinants of preterm neonatal mortality admitted to WSUCSH, with some variations from other studies carried out in in terms of the environment, hospital setting and the inclusion of additional factors.

### Methods

An institution-based, unmatched case-control study was conducted from June 1–15, 2023 on preterm neonates who admitted to Wolaita Sodo University Comprehensive Specialized Hospital from July 1, 2020, to June 30, 2022. A total of 690 with 1:2 ratio (230 cases and 460 controls respectively) preterm neonate medical charts were used for data extraction using a pre-established tool. Data were checked for completeness and consistency, entered using Epi-Data V.4.6, and transported to SPSS version 26 for analysis. Multivariate logistic regression analysis was performed to identify the independent determinants of preterm neonate mortality at a p value of < 0.05 along with a 95% confidence interval (CI). Finally, the model fitness of the study was assessed using the Hosmer and Lemshow model fitness test.

sensitive patient information (Even de-identified). Data are available from the authors after appropriate request to the School of Public Health Research Ethics Committee, Wolaita Sodo University Email contact: wsuniv@ethionet.com for researchers who meet the criteria for access to confidential data.

**Funding:** The author(s) received no specific funding for this work.

**Competing interests:** The authors have declared that no competing interests exist.

**Abbreviations:** ANC, Antenatal Care; AOR, Adjusted Odds Ratio; APGAR, Appearance, Pulse, Grimace, Activity, and Respiration; COR, Crude Odds Ratio; CPAP, Continuous Positive Airway Pressure; EDHS, Ethiopian Demographic and Health Survey; FMOH, Federal Ministry of Health; HMD, Hyaline Membrane Disease; KMC, Kangaroo Mother Care; NICU, Neonatal intensive care unit; PPROM, Preterm premature rupture of membrane; RDS, Respiratory Distress Syndrome; WHO, World health organization; WSUCSH, Wolaita Sodo University Comprehensive Specialized Hospita.

## Results

The preterm neonatal death rate was 10.9%. Hypothermia (AOR = 1.66, 95% CI:1.03–2.67), Sepsis (AOR = 3.35, 95% CI: 1.25 8.96), hypoglycemia (AOR = 2.85, 95% CI:1.09–7.44), Respiratory distress syndrome(RDS) (AOR = 7.21 95% CI:2.18–23.92), necrotizing enter colitis (AOR = 7.92, 95% CI:2.96–21.12) and neonates who stayed at hospital less than 7 days (AOR = 7.36, 95% CI:2.82,19.22) were predictors of preterm mortality.

## Conclusion and recommendation

Preterm neonatal mortality in this setting is mainly related to sepsis, respiratory distress syndrome (RDS), necrotizing enter colitis (NEC), hypoglycemia, hypothermia, and brief hospital stay. Thus, it would be preferable to offer appropriate prevention measures and extra care for neonates who have those factors.

## Introduction

Neonatal mortality is a core indicator of neonatal health defined as death during the first 28 days of life [1]. The World Health Organization (WHO) defines a preterm birth neonate as a neonate who was born before 37 full weeks of pregnancy. Extremely preterm neonates (28 weeks), very preterm neonates (28–32 weeks), and moderately preterm neonates (32–37 weeks) of gestation are additional categories for preterm neonates [2, 3]. Even though prematurity is a global burden, there is survival difference between developing and developed countries. Neonates born in Africa had 12 times higher risk of mortality compared to those born in Europe [4].

Approximately 45% of the deaths from preterm delivery difficulties happened within 48 hours, and 73% happened during the first week. Infection (23%), preterm labor (34%) and birth asphyxia (26%), according to reports, are the leading causes of neonatal mortality worldwide [5].

The leading cause of death for children under the age of five was preterm birth complications (1.055 million [95% CI, 0·935–1·179]) [6]. Particularly those in Southeast Asia and sub-Saharan Africa, low- and middle-income nations bear a disproportionately heavy burden of preterm birth. In many nations, preterm birth rates are rising. For the United Nations Sustainable Development Goal 3 goal #3.2, which aims to eliminate all preventable deaths of newborns and children under the age of five by 2030, the issue of premature birth is of utmost significance [4].

Ethiopia is one of the top five countries where almost half of all the global neonatal mortality occurred. While the world has targeted to reduce neonatal mortality to at least 12 per 1,000 live births by 2030, Ethiopia is still recording persistently high neonatal mortality. The Ethiopian Mini Demographic and Health Surveys 2019 reported that Neonatal mortality declined from 39 deaths per 1,000 live births in 2005 to 29 deaths per 1,000 live births in 2016 before increasing to 33 deaths per 1,000 births in 2019 (an overall reduction of 15% over the past 14 years) reflects that Ethiopia is far behind the goal set at the national level which was reducing neonatal mortality to 10 per 1, 000 live births by 2020, [7, 8] and according to report of United Nations of children fund, preterm birth which accounts 23% was believed to be a major and direct cause of neonatal death [2, 9].

According to a report by the Federal Ministry of Health of Ethiopia and different studies, preterm birth is the first cause of neonatal mortality and the fourth cause of under-5 mortality [8, 10]. Additionally, determinants are not well recognized in Ethiopia primarily at the institutional level or in the study area, but few studies have reported causes such as sepsis, asphyxia, HMD and congenital malformations [11–13].

Neonatal gestational age less than 37 weeks, low birth weight less than 2,500 grams, an Apgar score of less than 7 at 5 minutes, and referral neonates correlated with neonatal mortality in Democratic Republic of the Congo [14]. ANC follow-up, multiple pregnancies, preeclampsia, HIV/AIDS, anemia, and age 34 years are among the maternal risk factors [10]; where as, pregnancy-induced hypertension, a protracted membrane rupture, low birth weight, and very low birth weight were the main contributing factors in other study in Ethiopia [15].

Hypothermia (67.2%), RDS (43.0%), SGA (15.7%), and perinatal asphyxia (14.5%) were found to be the most common medical problems among preterm neonates in Uganda, but in studies done in Nigeria, maternal medical illnesses had a detrimental impact on newborn survival [16, 17]. Furthermore, immaturity-related (26%), perinatal asphyxia (26%), infection (23%), associated with neonatal death [15].

Neonatal mortality has dropped in Sub-Saharan Africa during the past 20 years, including Ethiopia, but not as much as was anticipated. Although the overall infant mortality rate in this study environment was 27 per 1000 neonates, [18] the rate and contributing factors of preterm neonatal mortality were not previously well understood. To the best of our knowledge, limited studies have been performed in our country, Ethiopia; regarding determinants of preterm mortality. However, those studies did not agree on specific predictors in different health care settings. However, most of the causes of death can be manageable and preventable. Finding the specific predictors, might be a crucial step towards lowering infant mortality in study setting and other low income countries. Additionally, as mentioned in the methods section, the research environment is different from other similar study locations in terms of the availability of medical equipment and medical specialists as well as the inclusion of other factors that may contribute to mortality (e.g., pregnancy care: induction of delivery, medical disorders: thyroid disorder; newborn characteristics: weight to GA and apnea of prematurity, length of hospital stay (LOS) were taken into consideration; LOS was one of the significant factors according to this study in addition to others). Therefore, this study aimed to identify the determinants of preterm neonatal mortality in neonatal intensive care units (NICUs) of the Wolaita Sodo University Specialized Hospital, Southern Ethiopia.

## Methods

### Study design

A facility-based unmatched case–control study was employed

### Study setting and period

This study was conducted at Wolaita Sodo University Comprehensive Specialized Hospital from June 1–15, 2023. The WSUCSH is located in Sodo town, the capital of the Wolaita Zone, about 200.81 miles or 328 kilometers south of Ethiopia's capital, Addis Ababa. It has a latitude and longitude of 6°54′N 37°45′E with an elevation between 1501 to 3000 meters above sea level and the climate Sodo has a bi-modal rainfall pattern that extends from March to October. The first rainy period occurs in March to May, while the second rainy period covers July to October, with its peak in July/August, receiving 169.5 cm of rain annually. Based on the 2018 Population Projection by the CSA, this town has a total population of 254,294, of whom 125,855 are men and 128,439 are women. There are 28 high and medium private clinics, four

private hospitals, five health centers, and one Comprehensive Specialized Hospital in Sodo town. There are roughly 156 educational institutions in Sodo Town overall (63 public and 93 private). In addition to one university, there are five colleges, five secondary and preparatory schools, seven international hotels, and seven road networks that connect the various Woreda's in the area and the surrounding areas. Cereals, roots, tubers, and vegetables are common staple foods in the area (Wolaita Zone Health Department Annual report and Sodo city administration office). Currently the WSUCSH has 370 beds with 1760 health professionals and 568 non-health (supportive) staff serving approximately more than 5 million people in Wolaita and neighboring zones of catchment population. As per clinical service, it provides outpatient, emergency and inpatient service in all clinical departments namely internal medicine, general surgery and orthopedics, gynecology and obstetrics, pediatrics and ophthalmology. It also has dermatology, dental and psychiatric service.

Neonatal Intensive care Unit is one of Pediatric and child health department health service providing unit as like as Pediatric emergency, Pediatric outpatient, pediatric inpatient, Pediatric ICU and Nutritional rehabilitation unit within the department. NICU have Term and Preterm isolation, KMC, Septic Ward and Mothers waiting rooms.

The pediatrics and child health department employs nine pediatricians, twenty-three residents, four general practitioners, and six to eight medical interns who rotate monthly between the NICU and other departmental units. Of these, one pediatrician, four residents, one general practitioner, and two to three medical interns work in the NICU each month and 17 trained nurses with patient to nurse ratio of six to one respectively. NICU have 30 functional beds with average admission of 110 per month. For newborns in need of high care, particularly surfactant and ventilation, High Care provides NCPAP but does not provide therapeutic hypothermia, invasive mechanical ventilation, or surfactant services. Instead, patients are sent to the local private health sector.

## Participants

The source population comprised all preterm neonates admitted to Wolaita Sodo University Comprehensive Specialized Hospital between July 1, 2020, and June 30, 2022; the study was conducted from June 1–15, 2023: the study population consisted of all preterm neonates hospitalized between July 1, 2020, and June 30, 2022, with a diagnosis of prematurity; additionally, preterm neonates who died within the first 28 days of life and were verified by a physician were considered cases; 2,259 preterm newborns were able to live for 28 days in total. 460 newborns were chosen as controls from among them by taking into account that, for every case, two alive controls were chosen from the registry book of admission based on roughly born on the same date as the cases (-/+ 2 days) [19].

## Inclusion and exclusion criteria

The study included preterm babies aged less than or equal to 28 days and whose charts contained comprehensive medical information about issues relating to date of admission, diagnosis, treatment, and discharge outcome (dead or alive).

Recorded data of preterm babies with no clear end outcome or referrals or those who were early self-discharged against medical advice and neonates with lethal congenital malformations were excluded.

## Sample size determination and sampling procedure

Using online open-epi version 3 software with a case-to-control ratio of 1:2, a power of 80%, and a confidence interval of 95%, we used previously conducted similar type studies relevant variables to determine the sample size for this investigation. The variables GA at delivery <28

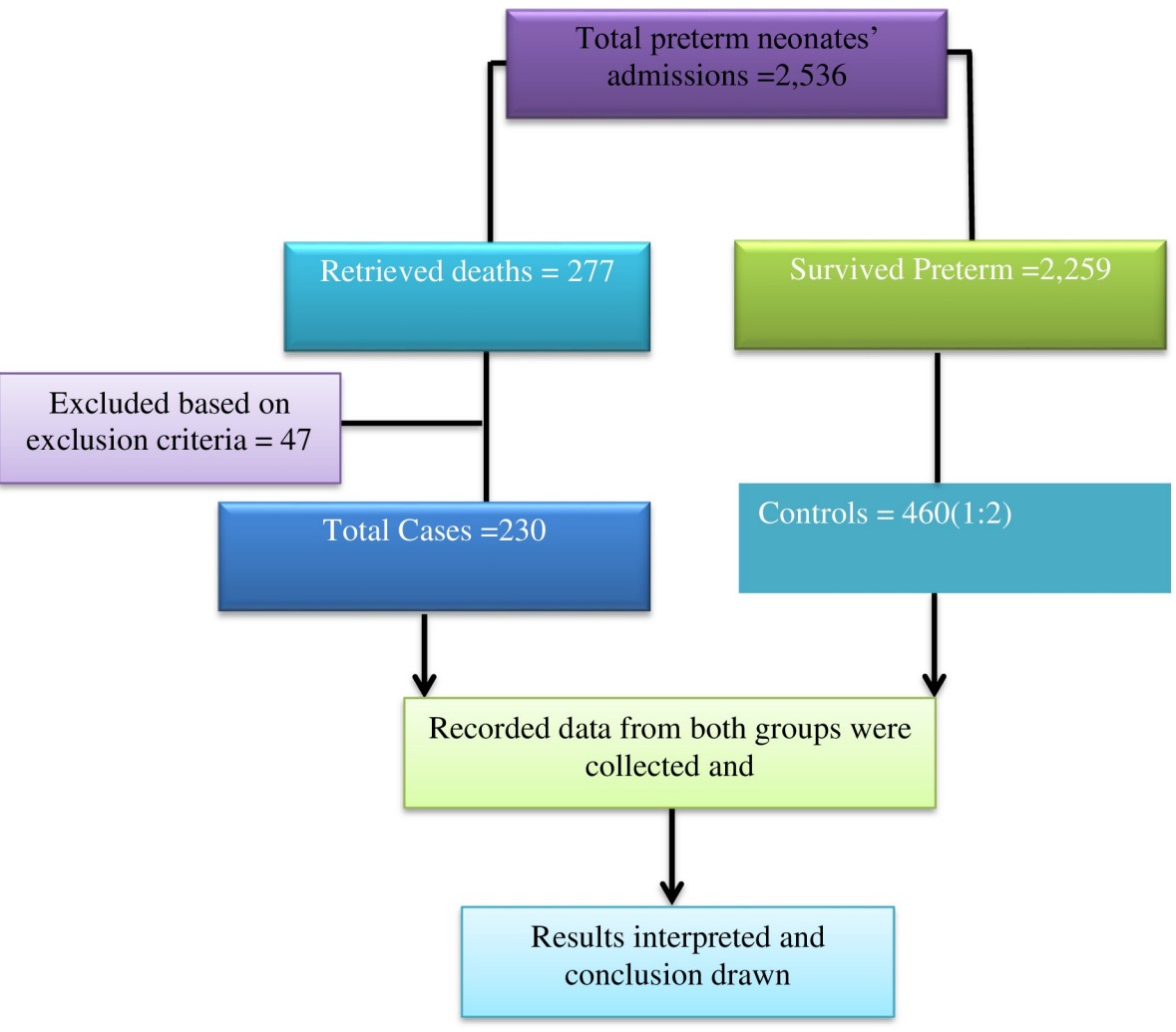

**Fig 1. Flow diagram depicting preterm neonate charts enrolled in a study with prematurity at Wolaita Sodo University Comprehensive Specialized Hospital between July1, 2020 and June 30, 2022.**

weeks; n = 165, 1min APGAR: n = 216, Birth weight 1.5–2.49 Kg: n = 218; Maternal DM: n = 420; and preterm neonatal sepsis: n = 690 were obtained from a variety of published publications and arranged in increasing order of sample size result [10, 20]. As result, the variable of preterm neonatal sepsis had the greatest sample size, 690, with a case-to-control ratio of 1:2 (230 cases and 460 controls), which was chosen as the study's final sample size. Using the inclusion criteria, the sampling frame was prepared for cases and controls from a serial list of medical record numbers from integrated admission and discharge logbooks in the NICU. Then, 690 preterm neonates medical record numbers were taken from the hospital archive room. The medical record numbers of charts were selected using the procedure in (Fig 1).

## Study variables

**Dependent variables.** Preterm Neonatal Mortality.

**Independent variables.** *Pregnancy care factors.* ANC, Induction of delivery, Mode of delivery. Place of delivery, Birth type.

*Medical disorders of the mother.* Hypertension, Diabetes mellitus (DM), and HIV/AIDS.

*Newborn characteristics*. Gestational Age, Neonatal Age at Admission. Birth Weight, 5th minute APGAR score, Apnea of Prematurity, KMC, Hypoglycemia, Necrotizing entercolitis, and Hypothermia.

**Data collection procedure and data quality management.** Ethiopian Public Health Institute (EPHI) facility-based perinatal death surveillance data abstraction format with 8 parts was adapted [21] and modified. Then, the abstraction form was designed in 4 parts: general information of the neonate and the mother, Maternal medical illness related, Pregnancy related factors and neonatal related factors. In addition to one public health expert supervisor, three BSc nursing professionals and two medical interns who were working in the NICU, had a solid grasp of medical terminology, and were acquainted with NICU documentation on patient charts and registries were assigned as data collectors on a part-time basis. Before the actual data collection started, one day of training was given to data collectors and supervisors on how to collect and record data appropriately. The data abstraction tool was pretested with 5% of the sample size, and slight modifications were made. Based on registration numbers, all charts were identified by the study team and provided for data collectors. Then, from all 690 (230 cases and 460 controls) charts, relevant patient data were extracted by data collectors.

**Data analysis procedures.** The collected data were checked for completeness and consistency and cleaned, then entered into Epi-Data version 4.6, and then transferred to SPSS version 26 for analysis. Descriptive statistical analysis, such as frequency and proportion, was used to describe the study participants. Binary logistic regression analysis was carried out, and the candidate variables were selected at a p value of 0.25 for multivariate logistic regression. Multiple logistic regression was performed, and independent determinants of preterm neonates mortality were identified at a p value of 0.05 along with a 95% CI. In the final multivariate models, the level of multicollinearity was checked and fitted using variance inflation factor (VIF) and tolerance and found within a tolerable range: all variable values $<5$ and $> 0.1$ &$<1$ respectively. Meanwhile, the model fitness of the study was assessed using the Hosmer and Lemshow model fitness test with test result of (Pvalue = 0.77, Negelkerke R2 = 57.9%). The results were finally presented using text, tables and figures.

## Ethics approval and consent to participate

Ethical clearance (Ref: No. CRCSD 4/3719/2011) was obtained from the research and ethics committee (REC) of the School of Public Health, College of Health Sciences, Wolaita Sodo University. Permission to access patients' records was granted by hospital officials. Patient information was anonymous and kept confidential. As we were reporting a retrospective study of medical records, informed consent was not required.

## Results

A total of 3,536 preterm neonates were admitted to the Neonatal Intensive Care Unit (NICU) at WSUCSH from 1st July, 2020 and 30th June, 2022 (three years), with a diagnosis of prematurity. Out of them 277 preterm neonates were died and 3,259 were discharged alive. According to the finding of this study, preterm neonatal mortality rate was 10.9%. Among these 690 preterm neonates charts (230 cases and 460 controls) were reviewed during the study period. Therefore, the data collection, analysis and interpretation were done for a total of 690 (230 cases and 460 control) neonate charts (Fig 2).

## Maternal and pregnancy related characteristics

Among 690(230 cases and 460 control) charts enrolled in the study; the majority (64.8%) of cases (deaths) were delivered from mothers between 20–34 years of age (Fig 2). More than half

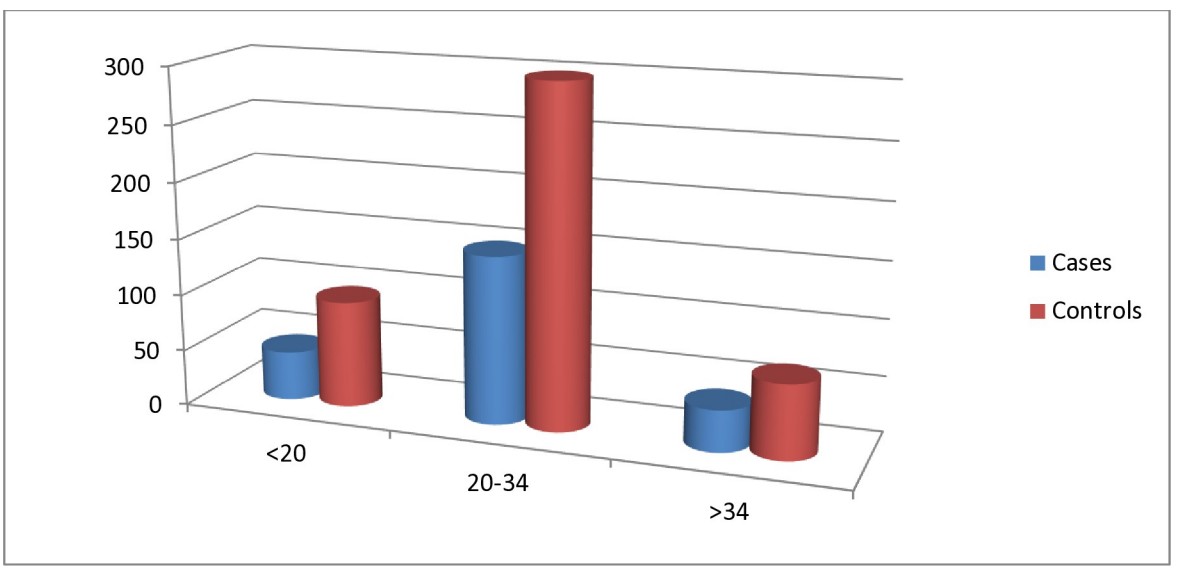

**Fig 2. Bar graph showing maternal age characteristics of the preterm neonates admitted to neonatal intensive care unit of Wolaita Sodo University Comprehensive Specialized Hospital from 1st July,2020 to 30th June, 2022.**

(51.3%) of deaths were born from mothers of multigravida (II-IV). From preterm neonates delivered from mothers who having medical cases majority delivered from mothers with pre-eclampsia 26(65%), however HIV, DM, Thyroid disease, (6(15%), 3(8%), and, 5(12%)) respectively (see Table 1).

## Neonatal factors

More than half 124(53.9%) cases and 258(56.1%) controls were born at the gestational age between 33 and 37 weeks while more than one third 96(41.8%) cases and 178(38.7%) of controls born between 28-32weeks of gestational age. Majority of preterm neonate cases 150 (65.2%) and controls 321(69.8%) were weighed between 1000–1499 gram however few of them (cases 18(7.8%) and controls 9(2.0%)) weighed less than 1000 grams (Fig 3). Hypothermia was reported from more than three fourth 174(76.5%) of cases and majority 285(62.0%) of controls while more than half 124(53.9%) of cases and about one-third 156(33.9%) of controls diagnosed with neonatal sepsis. Hypoglycemia reported from more than half 116(50.4%) of cases and one-fifth 96(20.9%) of controls. Approximately three-fourth 170(73.9%) of cases and majority 290(63.0%) of controls had respiratory distress syndrome (RDS) or Hyaline membrane disease (HMD). About one third 67(29.1%) of cases died in hospital within 24 hours and more than half 130(56.5%) of cases died within two to seven days (see Table 1).

## Determinants of preterm mortality

To choose potential variables for multivariate analysis, variables were first examined using bivariate analysis. Parity, birth weight of the neonate, hypothermia, neonatal sepsis, hypoglycemia, apnea of prematurity, respiratory distress syndrome, necrotizing enter colitis and hospital stay in days were candidate variables for multivariable analysis with a P-value of less than 0.25.

After potential confounders were taken into account in multivariate analysis, hypothermia, neonatal sepsis, hypoglycemia, respiratory distress syndrome, necrotizing enter colitis and hospital stay in days, were found to be determinants of an increased risk of preterm neonatal mortality.

**Table 1. Maternal, pregnancy and neonatal characteristics of the preterm neonates admitted to neonatal intensive care unit of Wolaita Sodo University Comprehensive Specialized Hospital from 1st July,2020 to 30th June, 2022.**

| Variables | Category | Cases:n (%) | Controls:n (%) | COR 95% CI | P-value |
|---|---|---|---|---|---|
| Residence | Urban | 147(63.9) | 173(37.6) | 1 | 0.697 |
| | Rural | 83(36.1) | 287(62.4) | 1.07(0.77,1.48) | |
| Parity | Primigravida | 85(37.0) | 180(39.1) | 1 | 0.239* |
| | Multigravida | 118(51.3) | 239(52.0) | 0.72(0.41,1.24) | |
| | Grandmultigravida | 27(11.7) | 41(8.9) | 0.75(0.44,1.28) | |
| ANC | Yes | 209(90.9) | 416(90.4) | 0.95(0.55,1.64) | 0.84 |
| | No | 21(9.1) | 44(9.6) | 1 | |
| Induced delivery | Yes | 37(15.2) | 68(14.6) | 0.94(0.60,1.45) | 0.519 |
| | No | 193(83.9) | 392(85.2) | 1 | |
| Mode of delivery | Spontaneous Vertex delivery | 169(73.5) | 342(74.3) | 1 | 0.76 |
| | Caesarian section | 56(24.3) | 107(23.3) | 0.94(0.65,1,37) | |
| | Instrumental | 5(2.2) | 11(2.4) | 0.87(0.29,2.62) | |
| Place of delivery | Health institution | 202(87.8) | 406(88.3) | 1 | 0.868 |
| | Home | 28(12.2) | 54(11.7) | 0.96(0.59,1.56) | |
| Medical disease | Yes | 40(17.4) | 139(30.2) | 0.97(0.69,1.37) | 0.861 |
| | No | 190(82.6) | 321(69.8) | 1 | |
| Type of disease | Preeclampsia | 26(65) | 72(69.2) | 0.85(0.64,1.38) | 0.73 |
| | Others(HIV,DM,Thyroid disorders) | 14(35) | 32(12.5) | 1 | |
| Gestational Age(in weeks) | ≤ 28 | 10(4.3) | 24(5.2) | 1 | 0.697 |
| | 28+1–32 | 96(41.8) | 178(38.7) | 1.2(0.50,3.02) | |
| | 32+1–37 | 124(53.9) | 258(56.1) | 1.2(0.45,3.10) | |
| Weight to Gestational age | SGA | 199(8.3) | 40(8.7) | 0.87(0.40,1.87) | 0.85 |
| | AGA | 211(91.7) | 420(91.3) | 1.12(0.80,1.56) | |
| | LGA | 0(0.0) | 0(0.0) | 1 | |
| APGAR score at 5minutes | < 3 | 38(16.5) | 73(15.9) | 1.12(0.68,1.97) | 0.8 |
| | 3–6 | 148(64.3) | 289(62.8) | 1.14(0.76,1.72) | |
| | ≥ 7 | 44(19.1) | 98(21.3) | 1 | |
| Hypothermia | Yes | 174(76.5) | 285(62.0) | 2.08(1.45,2.97) | <0.001* |
| | No | 54(23.0) | 175(38.0) | 1 | |
| Sepsis | Yes | 124(53.9) | 156(33.9) | 2.28(1.65,3.15) | <0.001* |
| | No | 106(46.1) | 304(66.1) | 1 | |
| Hypoglycemia | Yes | 116(50.4) | 96(20.9) | 3.86(2.74,5.44) | <0.001* |
| | No | 116(49.6) | 364(79.1) | 1 | |
| Jaundice | Yes | 74(32.2) | 143(31.1) | 0.95(0.68,1.34) | 0.772 |
| | No | 156(67.8) | 317(68.9) | 1 | |
| Apnea of Prematurity | Yes | 91(39.6) | 109(23.7) | 0.72(0.41,1.24) | 0.21* |
| | No | 139(60.4) | 351(76.3) | 1 | |
| KMC cared out | Yes | 37(16.1) | 112(24.3) | 1.68(0.92,2.53) | 0.82 |
| | No | 197(83.9) | 348(75.7) | 1 | |
| RDS/HMD | Yes | 170(73.9) | 290(63.0) | 1.66(1.17,2.36) | 0.04 * |
| | No | 60(26.1) | 170(37.0) | 1 | |
| PNA | Yes | 27(11.7) | 60(13.0) | 1.13(0.69,1.83) | 0.627 |
| | No | 203(88.3) | 400(87.0) | 1 | |

*(Continued)*

**Table 1.** (Continued)

| Variables | Category | Cases:n (%) | Controls:n (%) | COR 95% CI | P-value |
|---|---|---|---|---|---|
| Necrotizing Enter-colitis | Yes | 137(59.6) | 83(18.0) | 6.69(4.69,9.54) | <0.001* |
| | No | 93(40.4) | 377(82.0) | 1 | |
| Hospital stay in days | 1 | 67(29.3) | 4(0.8) | 34.4(16.40,72.1) | <0.001* |
| | 2–7 | 130(56.5) | 104(22.6) | 12.3(8.1,18.6) | |
| | >7 | 33(14.3) | 352(76.5) | 1 | |

NB

* significant (P < 0.25 by bivariate analysis or candidate variables for multivariate analysis); 1 = Reference; COR (95% CI) = Crude odds ratio with 95% confidence interval; HIV = Human Immunodeficiency Virus; DM = Diabetes Mellitus; PNA = Perinatal asphyxia; KMC = kangaroo mother care

The odds of death were 1.7 times more likely observed among cases who had hypothermia than controls (AOR = 1.66, 95% CI = 1.03–2.67); Meanwhile, controls with sepsis had 3.4 times less likely odds of death than cases (AOR = 3.35, 95% CI = 1.25–8.96).

The odds of preterm death were three times more likely among preterm neonates who had hypoglycemia as compared to those preterm neonates who had no hypoglycemia (AOR = 2.85, 95%CI = 1.09–7.44).

Among neonatal related determinants the odds of preterm neonatal death from respiratory distress was 7.2 times more likely for cases than controls (AOR = 7.21, 95%CI = 2.18–23.92).

The odds of preterm neonatal death from necrotizing enter colitis was 7.9 times more likely among cases than controls (AOR = 7.92, 95%CI = 2.96–21.12).

Similarly, initial 24 hours of preterm neonates admission to NICU was determining the odds of death more than 32 times in cases than controls (AOR = 32.43, 95% CI = 4.02–261.34),

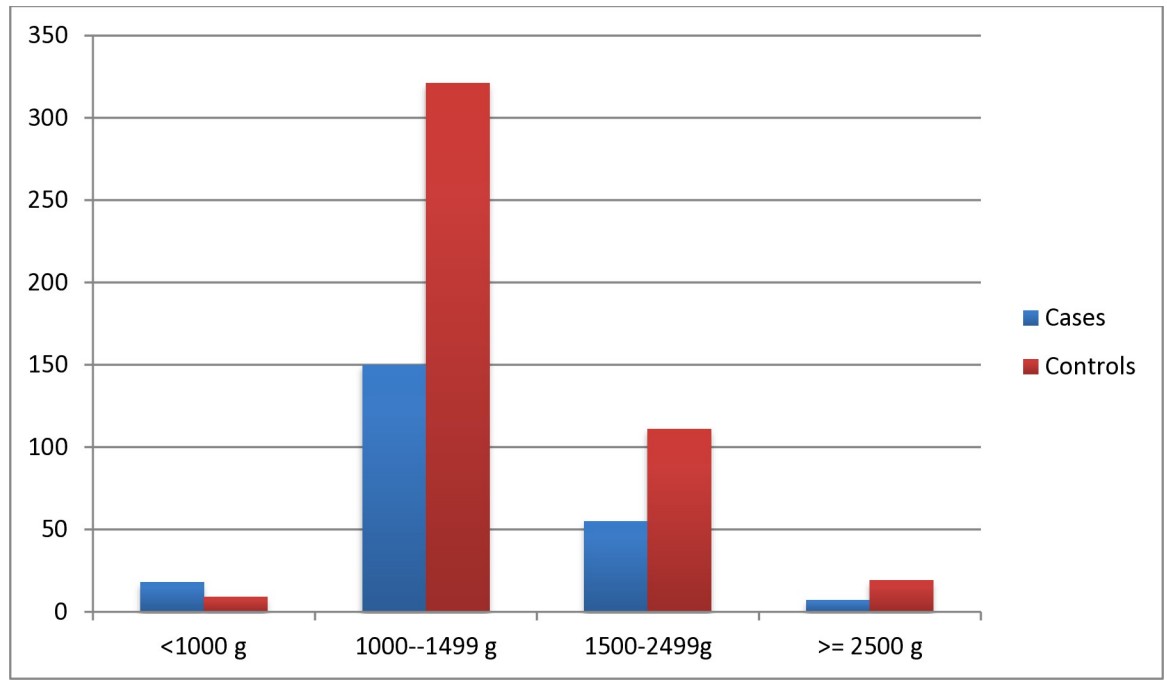

**Fig 3. Bar graph showing birth weight characteristics of the preterm neonates admitted to neonatal intensive care unit of Wolaita Sodo University Comprehensive Specialized Hospital from 1st July,2020 to 30th June, 2022.**

**Table 2. Determinants of preterm neonatal mortality among preterm neonates admitted to Neonatal Intensive Care Unit, Wolaita Sodo University Comprehensive Specialized Hospital from 1st July, 2020 to 30th June, 2022.**

| Predictor | Category | Cases n(%) | Controlsn(%) | COR(95%CI) | AOR(95%CI) | p-value |
|---|---|---|---|---|---|---|
| **Parity** | Primigravida | 85(37.0) | 180(39.1) | 1 | 1 | 0.109 |
| | Multigravida | 118(51.3) | 239(52.0) | 0.72(0.41,1.24) | 1.79(0.93,3.44) | |
| | Grandmulti gravida | 27(11.7) | 41(8.9) | 0.75(0.44,1.28) | 2.62(0.89,7.75) | |
| **Birth Weight in grams** | <1000 | 18(7.8) | 9(2.0) | 1 | 1 | 0.079 |
| | 1000–1499 | 55(23.9) | 111(24.1) | 5.43(1.67,17.66) | 0.25(0.05,1.17) | |
| | 1500–2499 | 150(65.2) | 321(69.8) | 1.34(0.53,3.39) | 0.18(0.04,1.78) | |
| | > = 2500 | 7(3.0) | 19(4.1) | 1.27(0.522,3.08) | 0.14(0.04,1.38) | |
| **Hypothermia** | Yes | 174(76.5) | 285(62.0) | 2.08(1.45,2.97) | 1.66(1.03,2.67) | 0.039* |
| | No | 54(23.0) | 175(38.0) | 1 | | |
| **Sepsis** | Yes | 124(53.9) | 156(33.9) | 2.28(1.65,3.15) | 3.35(1.25,8.96) | 0.016* |
| | No | 106(46.1) | 304(66.1) | 1 | | |
| **Hypoglycemia** | Yes | 116(50.4) | 96(20.9) | 3.86(2.74,5.44) | 2.85(1.09,7.44) | 0.033* |
| | No | 116(49.6) | 364(79.1) | 1 | | |
| **Apnea of Prematurity** | Yes | 91(39.6) | 109(23.7) | 0.72(0.41,1.24) | 0.53(0.26,1.09) | 0.09 |
| | No | 139(60.4) | 351(76.3) | 1 | | |
| **RDS/HMD** | Yes | 170(73.9) | 290(63.0) | 1.66(1.17,2.36) | 7.21(2.18,23.92) | 0.001* |
| | No | 60(26.1) | 170(37.0) | 1 | | |
| **Necrotizing Enter-colitis** | Yes | 137(59.6) | 83(18.0) | 6.69(4.69,9.54) | 7.92(2.96,21.12) | 0.001* |
| | No | 93(40.4) | 377(82.0) | 1 | | |
| **Hospital stay in days** | 1 | 67(29.3) | 4(0.8) | 34.43(16.40,72.1) | 32.4(4.02,261.) | 0.001* |
| | 2–7 | 130(56.5) | 104(22.6) | 12.3(8.1,18.6) | 7.7(2.82,19.22) | 0.001* |
| | >7 | 33(14.3) | 352(76.5) | 1 | | |

*NB*

*Significant (p<0.05), by multivariate logistic regression analyses; 1 = Reference; AOR (95% CI) = Adjusted odds ratio with 95% confidence interval; COR (95% CI) = Crude odds ratio with 95% confidence interval; RDS/HMD = Respiratory distress syndrome/Hyaline membrane disease

while, preterm neonates stayed for two to seven days had 7.7 times more likely odds of death for cases compared to controls (AOR = 7.7, 95% CI = 2.82, 19.22) (see Table 2).

## Discussion

In this facility based case-control study, the percentage of preterm newborn deaths was 10.9%, which is consistent with the 12.8% study carried out at Debretabor Town Health Institutions in Northwest Ethiopia [4]. This, however, is less than the results of a study done on preterm infants in various Ethiopian hospitals, where the percentage of premature deaths varied from 22.7 to 28.8% [4, 22]. A research by Muhe found that the overall percentage of premature deaths was 22.7%. However, the percentage of deaths in the NICU increased to 28.8% after removing patients who were not admitted. Thus, selection bias, care quality, and study regions may be related to the variation in the death rate observed in Ethiopian studies. Based on the results of this investigation, hypothermia, neonatal sepsis, hypoglycemia, RDS/HMD, necrotizing enter colitis, and short hospital stay were the major determinants of mortality in this study setup.

The study found that the odd of preterm neonatal death from sepsis was 3.4 times higher among cases than controls. This result is consistent with a study report from Adama Ethiopia, which found that sepsis-positive subjects had a 2.4-fold higher death rate than their counterparts [23]. This result is also consistent with a study conducted in MizanTepi, Ethiopia, which found that infants with sepsis had a greater mortality risk than those without the condition

[24]. The severe reaction to an infection, tissue damage, organ failure, and the absence of prompt medical attention could all contribute to this outcome [1].

Similarly, this study justifies the odds of premature newborn death was 7.2 times more likely among preterm babies with respiratory distress compared with their counterparts. This is consistent with studies done in Dessie Referral Hospital, Northeast Ethiopia and Iran abroad [15, 25] but higher in comparison to other study in Nigeria, Poland [6, 15]. The latter might be due to management difference of the settings related to health resource allocation including early surfactant administration and mechanical ventilation intervention to prevent respiratory failure before leading to death.

Current study reveals that preterm neonates that suffered by hypothermia had 1.7 times increased chance of neonatal death compared with no hypothermia. This is supported by studies done in Mizan Tepi, and Jimma University Hospital and Uganda [26–28]. This might be due to the fact that premature neonates are born with less adipose tissue which predisposes them to risk of complications of hypothermia, such as metabolic disturbance ending up with hypotension, electrolyte imbalance, metabolic acidosis, hypoglycemia and CNS insult and sepsis that might end-up with multi-organ failure causing death [29].

Furthermore, this facility based case-control study demonstrates significant association preterm death with necrotizing enterocolitis (NEC) which is the most frequent potentially fatal condition affecting the gastrointestinal (GI) tract in newborns. Thus, newborns with necrotizing enter colitis had a higher risk of premature neonate death than their counterparts. This is in line with studies done in selected Public hospitals in Addis Abeba, Ethiopia and the 2005 study conducted at the VON and in Johannesburg [30, 31]. In NEC, abdominal distention increases the risk of hypoxia and hypercapnia, which in turn causes apnea that requiring ventilation, hematologic and metabolic dysfunction that results in severe sepsis, and death from disseminated intravascular coagulopathy [29, 32].

This study also showed mortality of preterm neonates was higher among babies who stayed the first 24 hours compared with those stayed more than 7 days. These finding goes in line with other studies [9, 10, 17]. The first 7 days are the most critical period of a neonate's life which warrants close observation [17]. This is because of the period at which most of the intrauterine homeostasis will be changed to extra uterine adaptation almost all of the physiologic change are affected in preterm neonates; higher chance of having Patent ductus arteriosus which may lead to fluid overload and heart failure, Immature digestive system which might lead to NEC, RDS due to surfactant deficiency which will reach its nadir within 72 hrs, recurrent hypoglycemia due to decreased glycogen storage and gluconeogenesis because of immature preterm organs are some factors that contribute for early preterm deaths.

Furthermore, the poor health care setup such as having lack of surfactant and mechanical ventilation for management of RDS, neonatal sepsis, prolonged exposure to hypothermia and necrotizing entercolitis were contributors to preterm neonatal mortality. Thus, investigating the causes of preterm newborn death is intended to prevent preterm neonatal death.

## Limitations of study

The study was based on recorded data (chart review), so it's possible that some variables that weren't mentioned in the patient files were left out. For example, the absence of information regarding antenatal steroids, precautions taken against hypothermia, and patients who were released against medical advice were completely overlooked. The use of clinical diagnosis in place of investigation-based diagnostics, which may have been less reliable, may also have contributed to errors. One potential limitation of the study could be its retrospective methodology. Mortality may happen in certain cases even though the patient is discharged alive.

## Conclusions and recommendation

The preterm neonatal death rate of this setting was 10.9%. Neonatal infections, RDS, NEC, hypoglycemia, hypothermia, and brief hospital stays all have an impact on preterm death. Therefore, to reduce preterm neonatal mortality it would be crucial to provide adequate care and appropriate prevention measures for premature newborns identified with sepsis, hypothermia, RDS, NEC, hypoglycemia, and brief hospital stay (in the hospital for the first week of life).

## Operational definitions

**Appropriate for gestational age (AGA).**   The percentile of birth weight to gestational age is 10 and 90%) [33].

**Ballard score.**   Ballard system depends upon six physical and six neurologic criteria. The scores of each feature are added to calculate a maturity rating that correlates with gestational age and is accurate within two weeks [3].

**Cases.**   Preterm neonates who were died within 28 days after birth and had death summary.

**Controls.**   Preterm neonates who discharged as alive or passed 28 days of postnatal age and had discharge summary.

**Extremely low birth weight.**   Neonates born with less than 1000 g of birth weight [33].

**Extremely preterm.**   Neonates born less than 28 weeks of gestation [9].

**Feeding.**   Feeding stands for either trophic or full feeding.

**Large for gestational age (LGA).**   The percentile of birth weight to gestational age > 90%)

**Late preterm.**   Neonates born at 34 to 36 weeks and 6 days of gestation [9].

**Lethal congenital anomalies.**   Any congenital abnormal formation of fetal body parts that made the baby unable to survive.

**Low birth weight.**   neonates born with 1500–2499 g of birth weight [33].

**Moderate preterm.**   Neonates born at 32 to 33 weeks and 6 days of gestation [9].

**Normal birth weight.**   neonates with 2500–3999 g of birth weight [33].

**Preterm**, neonates who are diagnosed as preterm either by last normal mensuration period, by Ballard score or using early ultrasound (20 weeks) [33].

**Preterm neonatal mortality.**   Babies who were born at less than full 37 weeks of gestational age and died within 28 days of birthing [34].

**SGA.**   The percentile of birth weight to gestational age is less than10% [33].

**Very low birth weight.**   Neonates born with (1000–1499 g) of birth weight [33].

**Very preterm.**   Neonates born from 28 to 32 weeks of gestation [29].

## Acknowledgments

We feel grateful to Wolaita Sodo University, College of Health Sciences and Medicine for giving us the chance to conduct this research. Also, our heartfelt gratitude goes to data collectors and supervisors, Pediatrics department Neonatal Intensive Care nurses, HIMS office staff, and hospital medical archive room staff.

## Author Contributions

**Conceptualization:** Alemu Bogale, Ushula Deboch Borko, Solomon Abreha.

**Data curation:** Ushula Deboch Borko.

**Formal analysis:** Ushula Deboch Borko, Solomon Abreha.

**Methodology:** Alemu Bogale, Ushula Deboch Borko, Solomon Abreha, Bargude Balta.

**Resources:** Alemu Bogale.

**Supervision:** Alemu Bogale, Solomon Abreha.

**Writing – original draft:** Alemu Bogale, Ushula Deboch Borko, Bargude Balta.

**Writing – review & editing:** Bargude Balta.

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
