## [Decision Letter · Decision Letter 0]

8 Jul 2024

PONE-D-24-22309Determinants of mortality among preterm neonates admitted at Wolaita Sodo University Comprehensive Specialized Hospital, southern Ethiopia: An unmatched case-control studyPLOS ONE

Dear Dr. Bogale,

Thank you for submitting your manuscript to PLOS ONE. After careful consideration, we feel that it has merit but does not fully meet PLOS ONE’s publication criteria as it currently stands. Therefore, we invite you to submit a revised version of the manuscript that addresses the points raised during the review process.

We look forward to receiving your revised manuscript.

Kind regards,

Kahsu Gebrekidan, Ph.D.

Academic Editor

PLOS ONE

Journal Requirements:

2. For studies involving third-party data, we encourage authors to share any data specific to their analyses that they can legally distribute. PLOS recognizes, however, that authors may be using third-party data they do not have the rights to share. When third-party data cannot be publicly shared, authors must provide all information necessary for interested researchers to apply to gain access to the data. (https://journals.plos.org/plosone/s/data-availability#loc-acceptable-data-access-restrictions) 

Reviewers' comments:

Reviewer's Responses to Questions

**Comments to the Author**

1. Is the manuscript technically sound, and do the data support the conclusions?

Reviewer #1: Partly

Reviewer #2: Partly

2. Has the statistical analysis been performed appropriately and rigorously? 

Reviewer #1: Yes

Reviewer #2: No

3. Have the authors made all data underlying the findings in their manuscript fully available?

Reviewer #1: Yes

Reviewer #2: Yes

4. Is the manuscript presented in an intelligible fashion and written in standard English?

Reviewer #1: Yes

Reviewer #2: No

5. Review Comments to the Author

Reviewer #1: Dear authors!, the following points are my comments on your manuscript

_ In line #25, add a justification or rate neonatal death in your study area

_ In line #28 visited/admitted? please check your title and change to admitted

_ points in line # 184 #185 ( model fitness test), better if you add on line #32 as well with test result. In line #185, in addition to tables and figures for result presentation, add text

_ In line #34, add neonatal death rate from your finding before list of determinants

_ In line #39-44, i did not see any recommendation point raised. add recommendation based on your result

_In line #62, better if you write Ethiopian Mini Demographic...

_ When observing line # 62-63; and compare with statement written on line #81-82, what is your justification?

_ When I read statement written on line # 28-29 ,line #95 and line # 113-114, i did not understand your study period. please clarify it!

_ In line# 95-96, a term sodo or soho? 320 km or 328 km? check it!

_ In line #119 (-/+), do you have a reference?

_ In line # 128-132, I did not understand the statements. " to obtain a largest sample size, use online...". Please clarify.

_ In line # 157, operationalize your DV

_ In line # 192, put ( figure 1) in brackets

_ Add COR on Table 1 and 2; add AOR on Table 3

_ In line # 240-253, write your statements that regards to your determinants in a similar manner ( i.e. a number/percentage times more likely/less likely)

_ In line # 251 and others, your decimal number should be the same

_ In line # 259- 325 ( Discussion), you discussion and probable justification should be in line with those independent determinants

_ In line # 327-333 ( Limitation), how you overcome it?

_ Line # 336-342, revise your conclusion, and add your recommendation as per your finding

_ In line # 377 ( Ethical clearance letter/2011), and study period ( line # 95-96( June 1-15/2023). How you relate proposal preparation time with actual study period? (hint: retrospective case control study

Reviewer #2: General comments

Author should review the manuscript and correct some grammatical errors and the manuscript needs basic editing?

Line- 20, Abstract

Line-21, Background

Line 22-25. WHY not you give the definition of “Preterm birth” at introduction part of your manuscript?

Line-39, conclusion

Please it is better to add the word ‘recommendation” here

Line-46, introduction

Line,46-89, these sections did not provide a broad overview of neonatal death prevalence in your study area and its implementation in Ethiopia, it would be strengthened by adding more on the definitions of neonatal death and why it is important to study while determinants of neonatal death are well known before in many studies in Ethiopia?

Line-90, Methods

Line- 93, Study setting

Line 94-110; please add a map which, if possible, would also include the different Ethiopian locations presented in the discussion. Could the authors add some information on the geography of the city as well? Mountainous? Main activities? Seasons? Population??

Line- 186, Results

Line, 194-205, you described many texts but you also described these in table1. I recommend you to reduce the text?

Line 213-214, you stated as “But few of cases 10(4.3%) and controls 24(5.2%)

214 were born at below 28 weeks of gestational age” but how you get this data? If a mother gives birth less than 28 weeks of GA in Ethiopia, I will not considered as preterm rather abortion. Therefore, below 28 weeks of gestational age, it is considered as an abortion and could not be admitted? It needs clarification?

Line, 211-226- no need of many texts since it is described in Table 2?

Line, 227 (Table-2)- Categories of your variables specifically gestational age as (Gestational Age <=28 and Gestational Age 28-32) lacks clarity? At which categories you will classify GA 28 weeks?

Line, 230-253- What new variables or what new things you add in your study? Determinant factors for preterm neonatal death you described are already known and written in many books. What new result added by your study?

Line -258, Discussions

Line 259-325- unclear discussions. The discussion tries to compare and contrast the findings from other research findings which are not similar with the current study (in the context of study population, setting… Justification of the discrepancy is not as such strong? Please make it clear and precise

Line- 335, Conclusions

Better to say the heading “conclusions and recommendations” section

Line, 336-342-The study advised to improve prenatal care, refer high-risk pregnancies early, give timely prenatal steroids to mothers who may give birth before 34 weeks of gestational age, provide quality perinatal and neonatal care. Is there lack of prenatal care, is there a problem of refer high-risk pregnancies early and lack of give timely prenatal steroids to mothers who may give birth before 34 weeks? Did you assess these one? Recommendation should be based on your findings?

6. PLOS authors have the option to publish the peer review history of their article (what does this mean?). If published, this will include your full peer review and any attached files.

Reviewer #1: **Yes: **Tadele Amente, PhD candidate

Reviewer #2: **Yes: **Simachew Animen Bante

---

## [Author Response · Author response to Decision Letter 0]

16 Sep 2024

First and foremost, we would like to thank the academic editor for allowing us sufficient time to revise and address all of the reviewers' concerns. The reviewer's comments were critically evaluated, and necessary corrections or amendments were made. For comparison, we provided a response to the reviewer’s comment under each concern. We hope the academic editor and expert reviewers receive a significantly improved manuscript. Thank you incredibly much.!

Authors response for Reviewer 1

1.“ In line #25, add a justification or rate neonatal death in your study area.”

Response: Dear Reviewer, as you commented, we added justification as “Overall, the neonatal mortality incidence was 27 per 1000 neonates in study area (per a prior study published in BMC Pregnancy and Childbirth by Orsido et al.). The reason we used it is that this is the first study to show preterm mortality in this context, and there are no studies in the study area that show the mortality rate of preterms but rather the mortality of neonates in general. Thanks!

2. “In line #28 visited/admitted? please check your title and change to admitted.”

Response: Dear Reviewer, We have changed it to admitted. Thanks!

3. “points in line # 184 #185 ( model fitness test), better if you add on line #32 as well with test result. In line #185, in addition to tables and figures for result presentation, add text”

Response: Dear reviewer, we included it as written blow at main body of manscrupt methodology part. “In the final multivariate models, the level of multicollinearity was checked and fitted using variance inflation factor(VIF) and tolerance and found within a tolerable range: all variable values <5 and > 0.1 &<1 respectively. Meanwhile, the model fitness of the study was assessed using the Hosmer and Lemshow model fitness test with test result of (Pvalue=0.77, Negelkerke R2=57.9%). The results were finally presented using text, tables and figures. Thanks!

4. “In line #34, add neonatal death rate from your finding before list of determinants”

Response: Dear reviewer, your suggestion is appropriate: We have included it in result part and as well as in discussion. “According to this case control, the percentage of preterm newborn deaths was 10.9%”. Thanks!

5. “In line #39-44, i did not see any recommendation point raised. add recommendation based on your result.”

Response: Dear reviewer, we added recommendation in conclusion part with the sentence of: “Conclusion and Recommendation: Preterm neonatal mortality in this setting is mainly related to sepsis, respiratory distress syndrome (RDS), necrotizing enter colitis (NEC), hypoglycemia, hypothermia, and brief hospital stay. Thus, preterm mortality can be decreased by timely interventions such as quality perinatal and neonatal care, appropriate infection and hypothermia prevention measures, early initiation of trophic feeding and regularly check the serum sugar of preterm neonates.” Thanks!

6. “In line #62, better if you write Ethiopian Mini Demographic..”

.Response: dear reviewer, we re-wrote it as “The Ethiopian Mini Demographic and Health Surveys 2019’’.Thanks!

7. “When observing line # 62-63; and compare with statement written on line #81-82, what is your justification?”.

Response: dear reviewer, to make it has almost similar sense we re-write like this: ‘’The Ethiopian Mini Demographic and Health Surveys 2019 reported that Neonatal mortality declined from 39 deaths per 1,000 live births in 2005 to 29 deaths per 1,000 live births in 2016 before increasing to 33 deaths per 1,000 births in 2019 (an overall reduction of 15% over the past 14 years) (6).”. to indicate the decrement but it remain almost the same after 2016 E.C( not satisfying). Thanks!

6. “When I read statement written on line # 28-29 ,line #95 and line # 113-114, i did not understand your study period. please clarify it?” Response: dear reviewer, we apologize for this the mistake and we included the study period in respective areas as illustrated here: In Methods: An institution-based, unmatched case-control study was conducted from March 1 to 15, 2023 on preterm neonates who admitted to Wolaita Sodo University Comprehensive Specialized Hospital from July 1, 2020, to June 30, 2022.” Thanks!

7. “ In line# 95-96, a term sodo or soho? 320 km or 328 km? check it!?” 

Response: dear reviewer, we make it 328 km but there is different paths that may increase the distances but the shortest and easiest way is 328 Km. Thanks!

8. “In line #119 (-/+), do you have a reference?”

Response: dear reviewer, yes we have and we cited it. Thanks! 

9. ‘’In line # 128-132, I did not understand the statements. " to obtain a largest sample size, use online...". Please clarify:” 

Response: dear reviewer, We have “Sample size determination and sampling procedure: Using online open-epi version 3 software with a case-to-control ratio of 1:2, a power of 80%, and a confidence interval of 95%, we used previously conducted similar type studies relevant variables to determine the sample size for this investigation. The variables GA at delivery <28 weeks; n=165, 1min APGAR: n=216, Birth weight 1.5 -2.49 Kg: n=218; Maternal DM: n=420; and preterm neonatal sepsis: n=690 were obtained from a variety of published publications and arranged in increasing order of sample size result (8,17). As result, the variable of preterm neonatal sepsis had the greatest sample size, 690, with a case-to-control ratio of 1:2 (230 cases and 460 controls), which was chosen as the study's final sample size.. Thanks!

7. “In line # 157, operationalize your DV”

Response: dear reviewer, We operationalized it and cited with reference. “Preterm Neonatal Mortality:- Babies who were born at less than full 37 weeks of gestational age and died within 28 days of birthing’’. Thanks!

8. “In line # 192, put ( figure 1) in brackets.”

Response: dear reviewer, we did it.Thanks!

9. “Add COR on Table 1 and 2; add AOR on Table 3”

Response: dear reviewer, your comment is appropriate: We did according to your comment, one by one.Thanks!

10. “In line # 240-253, write your statements that regards to your determinants in a similar manner ( i.e. a number/percentage times more likely/less likely)”

Response: dear reviewer, as you suggested we narrated in the similar manner in the main body of manuscript. Like-“The odds of death were 1.7 times more likely observed among cases who had hypothermia than controls (AOR=1.66, 95% CI=1.03-2.67); Meanwhile, controls with sepsis had 3.4 times lower less likely odds of death than cases (AOR=3.4, 95% CI= 1.25-8.96).

The odds of preterm death were three times more likely among preterm neonates who had hypoglycemia as compared to those preterm neonates who had no hypoglycemia (AOR=2.8, 95%CI=1.09-7.44).

Among neonatal related determinants the odds of preterm neonatal death from respiratory distress was 7.2 times more likely for cases than controls (AOR= 7.2, 95%CI=2.18-23.92)…etc” Thanks!

11. “In line # 251 and others, your decimal number should be the same”

Response: dear reviewer, based on your right suggestion, we make decimals two throughout the findings. Thanks!

12. “In line # 259- 325 ( Discussion), you discussion and probable justification should be in line with those independent determinants?”

Response: kindly, reviewer you had the correct opinion. We partly re-wrote our discussion. Thanks!

13. “In line # 327-333 ( Limitation), how you overcome it?”

Response: kindly, reviewer we overcome some of limitations eg. The use of clinical diagnosis in place of investigation-based diagnostics, but which may have been less reliable, but still can be used in resource limited set-up. Regards!

14. “_ Line # 336-342, revise your conclusion, and add your recommendation as per your finding.”

Response:Please note that we include recommendation and modified our conclusion based on our finding only., “CONCLUSIONS AND RECOMMENDATION:-Neonatal infections, RDS, NEC, hypoglycemia, hypothermia, and brief hospital stays all have an impact on preterm death. To reduce preterm neonatal mortality, it is therefore advised to improve perinatal and neonatal care, take appropriate precautions against infection , hypothermia, RDS, NEC and hypoglycemia, and close observation of the baby in the hospital for the first week of life.” Thanks!

15. ‘’In line # 377 (Ethical clearance letter/2011), and study period ( line # 95-96( June 1-15/2023). How you relate proposal preparation time with actual study period? (hint: retrospective case control study’’

Response: kindly reviewer I apologize for the timing discrepancy. this is because the study's design changed from a planned cohort to a case-control, and it took longer to get in touch with the authors and the time of data collection extended to increase the sample size and ethical clearance was taken as it was. Thanks!

Reviewer 2

Authors response for Reviewer-2

“General comments: Author should review the manuscript and correct some grammatical errors and the manuscript needs basic editing?”

1. “Line- 20, Abstract and Line-21, Background?”

Response: kindly, reviewer, we re-wrote the Abstract part including background as illustrated on marked-up copy. Thus we added, neonatal mortality rate in study areas from previous studies, included study period in methodology part of abstract and statical model fitness that we used during analysis of this study. Thanks!

2. “Line 22-25. WHY not you give the definition of “Preterm birth” at introduction part of your manuscript?”.

Response: Dear reviewer, per your comment, we defined the preterm birth on the first line of introduction part as indicated here, “The World Health Organization (WHO) defines a preterm birth neonate as a child neonate who was born before 37 full weeks of pregnancy. Thanks!

3 . “Line-39, conclusion,Please it is better to add the word ‘recommendation” here?”

Response: kindly, reviewer you had the correct opinion. We added “Conclusion and Recommendation.” As shown here, 

Conclusion and Recommendation: Preterm neonatal mortality in this setting is mainly related to sepsis, respiratory distress syndrome (RDS), necrotizing enter colitis (NEC), hypoglycemia, hypothermia, and brief hospital stay. Thus, preterm mortality can be decreased by timely interventions such as appropriate infection, RDS, hypothermia, hypoglycemia and NEC prevention and treatment measures. Thanks!

4. “Line-46, introduction: Line,46-89, these sections did not provide a broad overview of neonatal death prevalence in your study area and its implementation in Ethiopia, it would be strengthened by adding more on the definitions of neonatal death and why it is important to study while determinants of neonatal death are well known before in many studies in Ethiopia?”

Response: dear reviewer; added definition of neonatal death, neonatal death prevalence of study area and current neonatal death rate and it implementation with comparison to SDG 2030 plan in the introduction part as follows:

# Neonatal mortality is a core indicator of neonatal health defined as death during the first 28 days of life (1)

#. Ethiopia is one of the top five countries where almost half of all the global neonatal mortality occurred. While the world has targeted to reduce neonatal mortality to at least 12 per 1,000 live births by 2030, Ethiopia is still recording persistently high neonatal mortality. The Ethiopian Mini Demographic and Health Surveys 2019 reported that Neonatal mortality declined from 39 deaths per 1,000 live births in 2005 to 29 deaths per 1,000 live births in 2016 before increasing to 33 deaths per 1,000 births in 2019 (an overall reduction of 15% over the past 14 years) reflects that Ethiopia is far behind the goal set at the national level which was reducing neonatal mortality to 10 per 1, 000 live births by 2020 (6,7)

#Overall, incidence of newborn mortality was 27 per 1000 neonates in this study setting, (10). Thanks!

5. “Line-90, Methods, Line- 93, Study setting, Line 94-110; 

please add a map which, if possible, would also include the different Ethiopian locations presented in the discussion. Could the authors add some information on the geography of the city as well? Mountainous? Main activities? Seasons? Population??;”

Response: dear reviewer; As you advised and indicated in the copy of the marked reviewed manuscript, we provided a map of the town that contained the study region and added some detail information about the city in addition to the hospital setting. “It has a latitude and longitude of 6°54′N 37°45′E with an elevation between 1501 to 3000 meters above sea level and The climate Sodo has a bi-modal rainfall pattern that extends from March to October. The first rainy period occurs in March to May, while the second rainy period covers July to October, with its peak in July/August, receiving 169.5 cm of rain annually. Based on the 2018 Population Projection by the CSA, this town has a total population of 254,294, of whom 125,855 are men and 128,439 are women. In Soddo town, there is one Comprehensive Specialized hospital, three private hospitals, five health centers, and twenty-eight high and medium private clinics. Cereals, roots, tubers, and vegetables are common staple foods in the area (Wolaita Zone Health Department Annual report). Thanks!

6. “Line- 186, Results, Line, 194-205, you described many texts but you also described these in table1. I recommend you to reduce the text?”

Response: dear reviewer; we reduced text part. Thanks!

7. “Line 213-214, you stated as “But few of cases 10(4.3%) and controls 24(5.2%) 214 were born at below 28 weeks of gestational age” but how you get this data? If a mother gives birth less than 28 weeks of GA in Ethiopia, I will not considered as preterm rather abortion. Therefore, below 28 weeks of gestational age, it is considered as an abortion and could not be admitted? It needs clarification?”

Response: Dear reviewer, you are right, In Ethiopia, a baby born before 28 weeks of gestation was considered an abortion. However, as of right now, the federal ministry of health is advising that newborns older than 26 weeks be admitted (a new guideline that is being implemented and trained upon), and that babies under 28 weeks should receive different care. For this reason, as far as I know, there have been cases in our hospital where neonates have improved and grown up with follow-up, even though their gestational age was less than 28 weeks. For this reason, neonates GA< 28 wks were also included. Thanks!

8. “Line, 211-226- no need of many texts since it is described in Table 2?.”

Response: dear reviewer, we reduced text part. Thanks!

9. “Line, 227 (Table-2)- Categories of your variables specifically gestational age as (Gestational Age <=28 and Gestational Age 28-32) lacks clarity? At which categories you will classify GA 28 weeks?”:

Response: Dear reviewer, we apologize for this dilemma and corrected it on the table 2. All neonates with complete gestational age of 28 weeks and less were categorized under < 28 weeks and those greater than 28 weeks(28 weeks + 1 days, were categorized under 28+1 -32 weeks). For this we reviewed studies (The Prevalence of Necrotizing Enterocolitis and Associated Factors Among Enteral Fed Preterm and Low Birth Weight Neonates Admitted in Selected Public Hospitals in Addis Ababa, Ethiopia: A Cross-sectional Study) that used this categorization before Thanks!

10. “Line, 230-253- What new variables or what new things you add in your study? Determinant factors for preterm neonatal death you described are already known and written in many books. What new result added by your study??”

Response: kindly, reviewer In addition to the variables listed in prior studies, we included a few variables (as marked yellow below) under each category that may be factors and provided evidence for our current finding that they had no association. 

To the best of the authors' knowledge, no other study has examined the rate of premature newborn mortality as opposed to neonatal mortality as a whole, hence this work offers this finding as a foundation for further investigations in this setup.

i. Distal level: those variables inherent to the mother and more unlikely to be altered;

ii. Intermediate level: those variables that may be affected by the distal level but at the same time may 

---

## [Decision Letter · Decision Letter 1]

17 Oct 2024

PONE-D-24-22309R1Determinants of mortality among preterm neonates admitted at Wolaita Sodo University Comprehensive Specialized Hospital, southern Ethiopia: An unmatched case-control studyPLOS ONE

Dear Dr. Bogale,

Thank you for submitting your manuscript to PLOS ONE. After careful consideration, we feel that it has merit but does not fully meet PLOS ONE’s publication criteria as it currently stands. Therefore, we invite you to submit a revised version of the manuscript that addresses the points raised during the review process.

We look forward to receiving your revised manuscript.

Kind regards,

Kahsu Gebrekidan, Ph.D.

Academic Editor

PLOS ONE

Journal Requirements:

Reviewers' comments:

Reviewer's Responses to Questions

**Comments to the Author**

1. If the authors have adequately addressed your comments raised in a previous round of review and you feel that this manuscript is now acceptable for publication, you may indicate that here to bypass the “Comments to the Author” section, enter your conflict of interest statement in the “Confidential to Editor” section, and submit your "Accept" recommendation.

Reviewer #1: All comments have been addressed

Reviewer #3: (No Response)

2. Is the manuscript technically sound, and do the data support the conclusions?

Reviewer #1: Partly

Reviewer #3: (No Response)

3. Has the statistical analysis been performed appropriately and rigorously? 

Reviewer #1: Yes

Reviewer #3: (No Response)

4. Have the authors made all data underlying the findings in their manuscript fully available?

Reviewer #1: Yes

Reviewer #3: (No Response)

5. Is the manuscript presented in an intelligible fashion and written in standard English?

Reviewer #1: Yes

Reviewer #3: (No Response)

6. Review Comments to the Author

Reviewer #1: Dear Author! your paper sound good and your revise every points raised earlier. Now, I have minor comment on the revised paper as follow:-

1. Line #111, in addition to health institution, better to add other infrastructure (i.e. educational institution, etc) in the town.

2. Line # 124, better if you explain/ add the mix of health professional with respective to specialty, sub-specialty working in NICU.

3. Line # 147 ' incomplete chart' is indirectly excluded. no need of write again as excluded.

4. Line # 182-183, from where you recruit the data collectors and supervisors; what is your criteria to select them as data collectors or supervisor?

5. For me, Table 1 and 2 are not important. Better if you use graph for either of the two.

Reviewer #3: The purpose of the research should be clearly stated in the abstract. The difference between this study and previous studies should be mentioned in the introduction (creativity and innovation).

7. PLOS authors have the option to publish the peer review history of their article (what does this mean?). If published, this will include your full peer review and any attached files.

Reviewer #1: **Yes: **Tadele Amente, PhD candidate

Reviewer #3: No

---

## [Author Response · Author response to Decision Letter 1]

30 Oct 2024

Authors response for Reviewer 1

1.“Line #111, in addition to health institution, better to add other infrastructure (i.e. educational institution, etc) in the town.”

Response: Dear Reviewer, per you comment, we added it. There are roughly 156 educational institutions in Sodo Town overall (63 public and 93 private). In addition to one university, there are five colleges, five secondary and preparatory schools, seven international hotels, and seven road networks that connect the various Woredas in the area and the surrounding areas. Thanks!

2. “Line # 124, better if you explain/ add the mix of health professional with respective to specialty, sub-specialty working in NICU.”

Response: Dear Reviewer, We have added it. As explained here: The pediatrics and child health department employs nine pediatricians, twenty-three residents, four general practitioners, and six to eight medical interns who rotate monthly between the NICU and other departmental units. Of these, one pediatrician, four residents, one general practitioner, and two to three medical interns work in the NICU each month and 17 trained nurses with patient to nurse ratio of six to one respectively.” Thanks!

3. “Line # 147 ' incomplete chart' is indirectly excluded. no need of write again as excluded.”

Response: Dear reviewer, we omitted it in main manuscript: Recorded data of preterm babies with no clear end outcome or incomplete chart or referrals or those who were early self-discharged against medical advice and neonates with lethal congenital malformations were excluded. Thanks!

4. “Line # 182-183, from where you recruit the data collectors and supervisors; what is your criteria to select them as data collectors or supervisor?”

Response: Dear reviewer, In order to address this issue, we attempted to rewrite our data quality assurance section in response to your query. The data collectors were nurses and medical interns who worked part-time in the NICU and were well-versed in medical terminology and documentation: In addition to one public health expert supervisor, three BSc nursing professionals and two medical interns who were working in the NICU, had a solid grasp of medical terminology, and were acquainted with NICU documentation on patient charts and registries were assigned as data collectors on a part-time basis. Thanks!!

5. “For me, Table 1 and 2 are not important. Better if you use graph for either of the two.”

Response: Dear reviewer, We reduced the number of tables by using graphs for some of the variables and combining the tables into a single table based on your feedback; however, it became difficult for us to display all of the variables (roughly 15-20) in a single table on graphs, so we combined tables 1 and 2 into a single table and used graphs for some of the variables, as shown in the main manuscript figure part.Thanks!

Authors response for Reviewer “Reviewer #3:

1. “The purpose of the research should be clearly stated in the abstract. The difference between this study and previous studies should be mentioned in the introduction (creativity and innovation)...”

Response: dear reviewer, based on your suggestion we tried to modify abstract in order to clarify the purpose of this study: as explained here, Although the general newborn mortality rate in the research area was 27 per 1000 neonates, but the preterm neonatal mortality rate and determinants were not previously well recognized. This study aimed to identify the Determinants of preterm neonatal mortality admitted to WSUCSH.

2) “The difference between this study and previous studies should be mentioned in the introduction (creativity and innovation)...”. Dear reviewer, We attempted to include what is novel in this study in comparison to earlier studies in introduction part based on your advice:- Additionally, as mentioned in the methods section, the research environment is different from other similar study locations in terms of the availability of medical equipment and medical specialists as well as the inclusion of other factors that may contribute to mortality (e.g., pregnancy care: induction of delivery, medical disorders: thyroid disorder; newborn characteristics: weight to GA and apnea of prematurity, length of hospital stay (LOS) were taken into consideration; LOS was one of the significant factors according to this study in addition to others). Thanks!

---

## [Editor Report · Decision Letter 2]

14 Nov 2024

Determinants of mortality among preterm neonates admitted at Wolaita Sodo University Comprehensive Specialized Hospital, southern Ethiopia: An unmatched case-control study

PONE-D-24-22309R2

Dear Mr. Alemu,

We’re pleased to inform you that your manuscript has been judged scientifically suitable for publication and will be formally accepted for publication once it meets all outstanding technical requirements.

Kind regards,

Kahsu Gebrekidan, Ph.D.

Academic Editor

PLOS ONE
---

## [Editor Report · Acceptance letter]

19 Nov 2024

PONE-D-24-22309R2 

PLOS ONE

Dear Dr. Bogale, 

I'm pleased to inform you that your manuscript has been deemed suitable for publication in PLOS ONE. Congratulations! Your manuscript is now being handed over to our production team.

Kind regards, 

on behalf of

Dr. Kahsu Gebrekidan 

Academic Editor

PLOS ONE